# *Stemphylium lycopersici* Nep1-like Protein (NLP) Is a Key Virulence Factor in Tomato Gray Leaf Spot Disease

**DOI:** 10.3390/jof8050518

**Published:** 2022-05-18

**Authors:** Jiajie Lian, Hongyu Han, Xizhan Chen, Qian Chen, Jiuhai Zhao, Chuanyou Li

**Affiliations:** 1State Key Laboratory of Crop Biology, College of Agriculture, Shandong Agricultural University, Tai’an 271018, China; lianjiajie123@163.com (J.L.); shanxihanhongyu@126.com (H.H.); c15650220616@163.com (X.C.); chenqiangenetics@163.com (Q.C.); 2Key Laboratory of Soybean Molecular Design Breeding, Northeast Institute of Geography and Agroecology, Chinese Academy of Sciences, Changchun 130102, China; 3University of Chinese Academy of Sciences, Beijing 100864, China; 4State Key Laboratory of Plant Genomics, National Centre for Plant Gene Research (Beijing), Institute of Genetics and Developmental Biology, Chinese Academy of Sciences, Beijing 100101, China

**Keywords:** *Stemphylium lycopersici*, NLP, tomato, grey leaf spot disease

## Abstract

The fungus *Stemphylium lycopersici* (*S. lycopersici*) is an economically important plant pathogen that causes grey leaf spot disease in tomato. However, functional genomic studies in *S. lycopersici* are lacking, and the factors influencing its pathogenicity remain largely unknown. Here, we present the first example of genetic transformation and targeted gene replacement in *S. lycopersici*. We functionally analyzed the *NLP* gene, which encodes a necrosis- and ethylene-inducing peptide 1 (Nep1)-like protein (NLP). We found that targeted disruption of the *NLP* gene in *S. lycopersici* significantly compromised its virulence on tomato. Moreover, our data suggest that NLP affects *S. lycopersici* conidiospore production and weakly affects its adaptation to osmotic and oxidative stress. Interestingly, we found that NLP suppressed the production of reactive oxygen species (ROS) in tomato leaves during *S. lycopersici* infection. Further, expressing the fungal NLP in tomato resulted in constitutive transcription of immune-responsive genes and inhibited plant growth. Through gene manipulation, we demonstrated the function of NLP in *S. lycopersici* virulence and development. Our work provides a paradigm for functional genomics studies in a non-model fungal pathogen system.

## 1. Introduction

The fungal genus *Stemphylium* includes many plant pathogens that infect economically important crops worldwide [1,2,3,4,5]. Four *Stemphylium* species, *S. lycopersici*, *Stemphylium solani*, *Stemphylium vesicarium*, and *Stemphylium botryosum* f. sp. *lycopersici*, are necrotrophic fungi that cause tomato (*Solanum lycopersicum*) gray leaf spot disease (GLS) [6]. GLS mainly attacks the foliage of tomato and forms gray lesions. Severe damage can lead to complete defoliation, making GLS one of the most damaging diseases of tomato [6,7]. However, the mechanism underlying the interaction between the plants and *Stemphylium* spp. is still largely unknown.

Plant-phytopathogen coevolution has driven the development of complicated mechanisms underlying plant-pathogen interaction. Plants recognize pathogen-associated molecular patterns (PAMPs)/microbe-associated molecular patterns (MAMPs) on the cell surface through pattern-recognition receptors (PRRs), which initiate PAMP-triggered immunity (PTI) [8]. To suppress host PTI and promote invasion, many pathogens secrete host-specific effectors [8,9]. In turn, plants become disease resistant by expressing resistance (R) proteins that perceive pathogen effectors and initiate effector-triggered immunity (ETI) [9]. Phytopathogens employ various approaches to facilitate successful infection, such as the secretion of cell wall degrading enzymes [10], toxins [11], and small RNAs [12].

Necrosis- and ethylene-inducing peptide 1 (Nep1)-like proteins (NLPs) comprise a large family of microbial secreted proteins with a broad taxonomic distribution that includes fungi, bacteria, and oomycetes [13,14,15]. NLPs are classed into three types based on protein alignment and phylogeny, among which type I is the most abundant and is present in fungi, bacteria, and oomycetes [15]. Type II NLPs have been identified only in fungi and bacteria and type III NLPs only in ascomycete fungi [15]. Toxic NLPs induce cell necrosis and trigger immune responses in dicotyledonous plants [9,16], whereas nontoxic NLPs do not induce cell necrosis but trigger immune responses [9,13,16].The central region of NLPs is conserved, containing a 20-amino-acid pattern (nlp20) or a 24-amino-acid pattern (nlp24) [17,18]. Studies in *Arabidopsis thaliana* (Arabidopsis) show that the toxic nlp20 triggers necrosis and immune responses resembling PTI [9], while the nontoxic nlp24 triggers PTI but not necrosis [13]. A leucine-rich repeat receptor protein, RECEPTOR LIKE PROTEIN 23 (RLP23), has been identified as the receptor for nlp20 in *Arabidopsis* [19].

Currently, GLS is commonly controlled by breeding resistant cultivars. In tomato, resistance to pathogenic *Stemphylium* spp. is controlled by the resistance gene *Sm*, which was introduced into cultivated tomato from its wild progenitor *Solanum pimpinellifolium* [6,20]. *Sm* encodes a nucleotide-binding site-leucine-rich repeat (NBS-LRR) resistance protein [21]. Since the virulence factor(s) in *Stemphylium* spp. are unknown, the tomato-*Stemphylium* interaction mechanism is also unclear. Necrotrophic phytopathogens use toxins and phytotoxic proteins as a central virulence strategy [22]. Metabolite profiling of *S. lycopersici* showed that the pathogen produces various phytotoxins, including the known compounds stemphylin and macrosporin [23,24], but no other virulence factors from *Stemphylium* spp. have been characterized. The *S. lycopersici* draft genome has been released and has provided the genetic information necessary for functional studies [25]. However, functional genomics studies in *Stemphylium* spp. have been lacking.

In this study, we identified and characterized the *NLP* gene from the *S. lycopersici* genome. To characterize its function, we carried out gene transformation and targeted gene replacement and generated *NLP* gene replacement mutants and overexpression strains. We demonstrated that NLP is a key virulence factor of *S. lycopersici* and triggers immune response in tomato. We also analyzed the role of NLP in fungal growth, conidiophore production, and stress adaption. Thus, we provide the first example of functional genomics in the non-model fungal pathogen *S. lycopersici*.

## 2. Materials and Methods

### 2.1. Plant Materials and Growth Conditions

Tomato (*Solanum lycopersicum*) cv. M82 was used as the wild type (WT) in this study. Following incubation in a 50 °C water bath for 30 min, seeds were sterilized by soaking in 2.7% sodium hypochlorite for 30 min. The sterilized seeds were rinsed three times with sterile water to remove excess sodium hypochlorite, evenly distributed on moist filter paper to stimulate germination, and left to germinate for 2–3 days. The plants were then transferred to compost soil mix (grass charcoal:vermiculite, 1:1) in a growth chamber at 24–26 °C and a relative humidity of 60%, with a white light intensity of 200 μmol photons m^−2^ s^−1^, and grown under a 16 h light/8 h dark photoperiod. For in vitro or in vivo infection experiments, 4-week-old plants were used [26].

### 2.2. Fungal Strain Isolation and Growth Conditions

The WT *S. lycopersici* strain was isolated from a tomato field in Beijing. Single spores were purified, and the progeny were stored. The WT strain was identified using the internal transcribed spacer (ITS) sequences ITS 1/4 [27], *gpd* and EF1α gene (Appendix A). To further confirm the WT strain is a *S. lycopersici* strain, we sequenced the strain and made a draft genome assembly based on second generation sequencing on Illumina platform (unpublished). The whole-genome Average Nucleotide Identity (ANI) was computed using FastANI (https://github.com/ParBLiSS/FastANI, accessed on 15 March 2022). The results showed that the ANI between the WT strain of *Stemphylium lycopersici* used and the released genome of strain CIDEFI 216 [25] was 99.3%, while the ANI between the WT strain and *Stemphylium vesicarium* (genome assembly GCA_008271585.1) was 89.8%, which suggests that the strain used in this study is a *S. lycopersici* strain.

The WT *S. lycopersici* strains were maintained at 25 °C in the dark on potato dextrose agar (PDA; Difco) and transferred weekly to fresh medium less than 3 generations. The frozen stocks of all strains were inoculated to reproduce conidia and the conidia suspension were used for experiment if needed. To test the effect of NLP on *S. lycopersici* adaption to ionic stress, conidia of the WT as well as ∆*nlp-1*, ∆*nlp-2*, *pACTIN:NLP-1*, and *pACTIN:NLP-2* strains (see below) were inoculated on (complete medium) CM medium (dextrose 10 g, tryptone 2 g, yeast extract 1 g, casein acids hydrolysate 1 g, NaNO_3_ 6 g, KCl 0.5 g, MgSO_4_·7H_2_O 1.0 g, KH_2_PO_4_ 1.5 g, agar 9 g, H_2_O to 1 L, pH 6.5) containing NaCl (1 M) or KCl (1 M) and incubated at 25 °C. After 8 days incubation the medium plates were measured for the area of the mycelium. CM medium without NaCl or KCl was used as the negative control. To test the effect of NLP on the adaption of *S. lycopersici* to oxidative stress, fungal cakes (5 mm in diameter) of WT, ∆*nlp-1*, ∆*nlp-2*, *pACTIN:NLP-1*, and *pACTIN:NLP-2* strains were inoculated on CM plates containing H_2_O_2_ (20 mM) [28] and incubated at 25 °C for 8 days. CM medium without H_2_O_2_ was used as the negative control.

### 2.3. Gene Knockout and Overexpression in S. lycopersici

∆*nlp-1* and ∆*nlp-2* strains were generated using a gene knockout method modified from split-marker and CRISPR strategies [29]. Primer information and vectors used for *NLP* knockout and overexpression experiments are shown in Appendix A. The targeted gene replacement vector harbors a hygromycin B resistance marker gene (*hph*). The PYF11-mCherry overexpression vector was constructed by replacing the green fluorescent protein (GFP) sequence in the PYF11 backbone with an mCherry sequence. *pACTIN:NLP-1*, and *pACTIN:NLP-2* strains were created in this vector using the *NLP* gene under the regulation of the 3-kb *S. lycopersici ACTIN* promoter, along with the first intron of the *ACTIN* gene [30].

### 2.4. Genetic Transformation of S. lycopersici

*S. lycopersici* genetic transformation was performed based on the method established by Connolly et al., with appropriate adjustments [31]. Spores were suspended in liquid medium and vortexed on ice. The concentration was adjusted to 1 × 10^5^ spores/mL, and 5 mL of the spore suspension was added to CM liquid medium and incubated for 12 h at 25 °C and 200 rpm. The mycelium (1 g wet weight) was digested by adding 20 mL of 0.7 M KCl buffer with 500 mg driselase (Sigma, St. Luis, Mo, USA, D8037), 100 mg lysing enzyme (Sigma, St. Luis, Mo, USA, L1412), 225 mg Cellulase R10 (Yakult, Tokyo, Japan, MX7352), and 45 mg Macerozyme (Yakult, Tokyo, Japan, MX7351) and incubated at 28 °C and 90 rpm. After digestion and filtering, protoplasts were harvested for transformation. The protoplast concentration was adjusted to 1 × 10^6^ protoplasts/mL. After washing twice with STC buffer (Sucrose 200 g, 0.5 M TRIS-HCl, 100 mL, CaCl_2_·2H_2_O 7.351 g, H_2_O to 1 L), 2–10 μg DNA fragments or plasmids to be transformed was added and mixed well, then 0.75 mL of PTC buffer (STC buffer containing 40% of PEG8000) was added. The mixture was incubated at room temperature for 30 min. Then regeneration medium (yeast extract 3 g, casein acids hydrolysate 3 g, sucrose 200 g, H_2_O to 1 L) was added and incubated at 28 °C and 200 rpm for 12 h. Targeted gene replacement or overexpression plasmids were introduced into protoplasts. Transformed protoplasts were then transferred to regeneration medium and cultured in screening medium with antibiotics. The knockout transformants were screened on regeneration medium containing 200 μg/mL hygromycin B. The putative transformants were first tested for deletion by NLP-TG-check primers (Appendix A) and further confirmed by the full-length of the target gene sequencing. The overexpression transformants were screened on regeneration medium containing 100 μg/mL G418. Putative transformants were first assessed for mCherry fluorescence and further confirmed by RT-qPCR. The fluorescence observation of mycelium was performed with a ZEISS^®^ LSM880 confocal microscope equipped with a 40× oil-immersion objective, and a 587 nm laser for mCherry fluorescence, controlled by ZEN Black software.

### 2.5. Bioinformatics Analysis and Phylogenetic Assay

The *Alternaria alternata* NLP (XP_018379771.1), *Botrytis cinerea* NLP1 (QIH49261.1) and NLP2 (XP_001551049.1) were selected as typical type I NLP. The *Pectobacterium parmentieri* NLP (WP_033071099.1) and *P. atrosepticum* (WP_011094611.1) NLP were selected as typical type II NLP. The *Aspergillus turcosus* (RHZ70022.1) NLP and *A. novofumigatus* NLP (XP_024685090.1) were selected as typical type III NLP. The sequences of *S. lycopersici* NLP and the typical type I, type II, and type III NLP proteins were analyzed by protein sequence alignment. *NLP* DNA and protein sequence data were obtained from NCBI (http://www.ncbi.nlm.nih.gov, accessed on 15 March 2022). A protein BLAST comparison was performed using fungal NLP protein sequences reported in previous studies [32]. Prediction of conserved domains was performed by blastp in the NCBI blast website. Sequence alignments and phylogenetic trees were generated using MEGA 11 software [33].

### 2.6. Tomato Disease Assay

Spores produced on sporulation medium were collected in a 50-mL centrifuge tube and vortexed with liquid CM medium for 1 min, and the suspension was filtered through Miracloth (Millipore, Cat: 475855, Burlington, MA, USA). The spores were collected by centrifugation at 4000 rpm and transferred into a new 50-mL centrifuge tube. Spores were washed and diluted to 1 × 10^5^ spores/mL with CM liquid medium. The disease assay was then performed based on a published method [26] with modifications. Two *S. lycopersici* spore suspensions (1 × 10^5^ spores/mL) were placed on each detached tomato leave, which was then placed on water agar medium (agar 9 g, H_2_O to 1 L). The Petri dishes were then sealed with permeable tape (3M, Cat: T8030C-0, St. Paul, MN, USA), and incubated for 5 days in a growth chamber under a 16 h light/8 h dark photoperiod. Disease symptoms were photographed, and lesion size was measured using ImageJ software [34]. 

### 2.7. DNA Constructs and Plant Transformation

DNA constructs for plant transformation were generated following standard molecular biology protocols and using Gateway (Invitrogen, Waltham, MA, USA) technology. The full-length *NLP* gene coding sequence was amplified by PCR from genomic DNA and cloned into the pK7FWG2 vector to generate *p35S:NLP.* The primers used to generate these DNA constructs are listed in Appendix A. The above constructs were introduced into tomato cv. M82 by *Agrobacterium tumefaciens*-mediated transformation [26]. Transformants were selected based on their resistance to kanamycin. Homozygous T_2_ or T_3_ transgenic plants were used for phenotypic and molecular characterization. 

### 2.8. ROS Staining 

Detection and quantification of ROS production during infection were performed as previously described [35,36]. Detached leaves from 4-week-old tomato plants were placed in Petri dishes containing 0.8% agar medium (agar dissolved in sterile water), with the petiole embedded in the medium. Tomato leaves infected with a 5-μL droplet of *S.*
*lycopersici* spore suspension at a concentration of 10^5^ spores/mL were collected 18 h after inoculation.

The infected leaves were stained with DAB (3,3′-diaminobenzidine, CAT: A03D10L104932) solution for H_2_O_2_ detection or with NBT (nitroblue tetrazolium chloride, CAT: A100329-0001) for O_2_^−^ detection. Detached leaves were submerged in DAB solution (1 mg/mL in 50 mM Tris-HCl, pH 3.8) or NBT solution (0.5 mg/mL NBT in 10 mM potassium phosphate buffer, pH 7.8). Vacuum infiltration was performed for 30 min, and leaves were stained for 4 h at 25 °C in the dark. The staining solution was then replaced with decolorization solution (acetic acid:glycerol:ethanol, 1:1:1, *v*/*v*/*v*) and the mixture boiled for 15 min. The decolored leaves were then scanned by scanner (EPSON perfection V330 Photo) and the scanned pictures were subjected to ImageJ for staining density quantification. The RGB values of scanned images were converted into a 16-bit grayscale, which were then quantified using the grayscale statistics method.

### 2.9. RNA Extraction and RT-qPCR

For gene expression quantification, samples were subjected to reverse transcription-quantitative PCR (RT-qPCR). Leaves of 2-week-old tomato plants were infected with a single 5-μL droplet of *S.*
*lycopersici* spore suspension (1 × 10^5^ spores/mL). The plants were then incubated in a growth chamber with high humidity. A droplet of CM liquid medium was used as the negative control. Infected leaves were harvested at different time points for RT-qPCR experiments.

For mycelium biomass quantification, detached leaves from 4-week-old tomato plants were placed in Petri dishes containing 0.8% agar medium, with the petiole embedded in the medium. Tomato leaves were infected with a 5-μL droplet of *S. lycopersici* spore suspension at a concentration of 10^5^ spores/mL. A droplet of CM liquid medium was used as the negative control. Infected leaves were harvested at 5 dpi for RT-qPCR experiments. Relative fungal growth was measured by RNA-based RT-qPCR using the threshold cycle value (*C_T_*) of *S. lycopersici ACTIN* against the *C_T_* of tomato *ACTIN2*.

Tomato seedlings were homogenized in liquid nitrogen and RNA was isolated using the total RNA kit (Cwbio, Cat: CW0559). First-strand cDNA was synthesized from 2-μg total RNA using the All-in-One First-Strand cDNA Synthesis Supermix for qPCR (One-Step gDNA Removal) kit (with DNase) (EasyScript, Cat: AE341-02) according to the manufacturer’s protocol. TransStart^®^ Green qPCR SuperMix was used for RT-qPCR reactions in the CFX Connect Real-Time System (Bio-Rad, Hercules, CA, USA). The RT-qPCR was performed by CFX96 Touch™ (Bio-Rad). Each reaction was performed in three biological replicates, with three technical replicates per biological replicate. Relative expression levels were calculated by the comparative 2^−ΔΔCt^ method [37]. Transcript abundance was based on the abundance of tomato Actin2. The primer sequences used in these experiments are described in detail in Appendix A. Error bars represent the standard error of the mean (SEM) from three biological replicates.

### 2.10. Quantification and Statistical Analysis

All experiments were repeated independently three times. At least four plants were included in each treatment in an independent experiment. Lesion area, fresh weight, spore concentration, plant height, and colony area data for quantification analyses are presented as average ± standard error (SE). RT-qPCR data for quantification analyses are presented as average ± standard error of mean (SEM). Statistical analyses were performed using routines implemented in DPS software. Comparisons among multiple groups were made by Duncan’s new multiple range test. Comparisons between two groups were made using Student’s *t* test (* *p* < 0.05, ** *p* < 0.01) with Microsoft Excel software.

## 3. Results

### 3.1. A Stemphylium lycopersici Type I Nep1-like Protein (NLP) Is Induced during Infection of Tomato

We subjected NLP protein sequences from *Botrytis cinerea*, *Fusarium graminearum*, and *Fusarium oxysporum* to a BLASTP search in the NCBI database. This identified only one copy of an NLP protein (GenBank accession: KNG44024.1) in the *S. lycopersici* genome [25]. This NLP protein sequence was predicted to have secretory signal peptides and an NPP1 (Necrosis-Inducing *Phytophthora* Protein 1) domain (Figure 1A). Phylogenetic analysis indicated that the NLP of *S. lycopersici* is a type I NLP protein (Figure 1B). Accordingly, multiple protein sequence alignment using all three NLP types from various species showed that the *S. lycopersici* NLP contains type I conserved sequences (Appendix A). Next, to assess *NLP* expression during host infection, we quantified the *NLP* gene transcription by Reverse Transcription-Quantitative PCR (RT-qPCR). Fungal conidia were inoculated on artificial medium as a negative control. RT-qPCR data showed that the *NLP* gene transcript abundance did not change on the control medium, but became significantly elevated in infected tomato leaves from 12 to 48 h post-inoculation (hpi; Figure 1C). These data suggest that the *NLP* gene expression is induced during *S. lycopersici* infection of tomato.

### 3.2. NLP Affects Conidiospore Production but Not Vegetative Growth in S. lycopersici

To study the molecular function of NLP in *S. lycopersici*, we first established a gene transformation method. Knockout of the *NLP* gene was conducted by targeted gene replacement through a CRISPR/Cas9 enhancement approach [29], in which Cas9 cut DNA and facilitated the targeted gene replacement. Using this method, we generated *NLP* gene knockout mutants (∆*nlp*) (Figure 2A). *NLP* gene deletion in screened transformants was confirmed by PCR (Appendix A), RT-qPCR (Appendix A), and gene sequencing (Appendix A). In addition, we constructed mCherry-tagged NLP protein under the regulation of the *ACTIN* promoter and first intron (*pACTIN:NLP*) (Figure 2B). RT-qPCR results showed high *NLP* gene expression levels in *pACTIN:NLP* strains (Appendix A). The *pACTIN:NLP* strains were subjected to microscopy and mCherry fluorescence was detected (Appendix A).

To investigate the role of NLP in fungal asexual reproduction, we quantified the conidia production of WT, ∆*nlp*, and *pACTIN:NLP* strains on sporulation medium. The results showed that *NLP* gene deletion reduced the ability of the ∆*nlp* mutant to produce conidia (Figure 2C). To test whether NLP regulates fungal development, we measured conidia germination and mycelial growth by inoculating WT, ∆*nlp*, and *pACTIN:NLP* strains in liquid complete medium (CM) and quantifying their germination after 6 h. The results showed no significant differences in spore germination rates between the tested strains (Appendix A, *p* value > 0.05). To compare mycelial growth in the different strains, we incubated conidia on CM medium and measured mycelium growth area after 8 days, finding that mycelial growth did not differ significantly among the tested strains (Figure 2D–G). We also measured the area of mycelium on CM medium at 2, 4, and 6 days after incubation to quantify the mycelium growth rate, and again found no statistically significant differences in rates between strains (Appendix A). These results indicate that *S. lycopersici* NLP does not influence either fungal conidia germination or vegetative growth. The results suggest that NLP affects conidia production but not the mycelium growth in *S. lycopersici*.

### 3.3. NLP Weakly Affects Osmotic and Oxidative Stress Adaptation in S. lycopersici

Infection-related stresses such as osmotic stress, oxidative stress, and cell wall integrity stress could be indictive of pathogen virulence [38]. To test whether NLP affects *S. lycopersici* adaptation to infection-related stresses [39], we compared the radial growth rates of WT, ∆*nlp*, and *pACTIN:NLP* strains on CM medium supplemented with osmotic stress agents. During incubation on CM medium with the osmotic stress agents NaCl (1 M) or KCl (1 M) for 8 days, osmotic stress significantly inhibited mycelium growth of the WT and both mutant strains, but whereas *pACTIN:NLP* showed a similar response to WT, the ∆*nlp* mutants showed a weakly yet significantly stronger growth inhibition (Figure 2D,E). To test whether *NLP* affects fungal adaptation to oxidative stress [28], we incubated the different strains on CM medium containing H_2_O_2_ (20 mM). Measurement of mycelium growth 8 days after inoculation again showed that the ∆*nlp* mutants exhibited weakly yet significantly stronger growth inhibition, while the *pACTIN:NLP* strains showed similar responses to the WT (Figure 2F,G). Next, to test whether NLP affects *S. lycopersici* adaptation to cell wall stability-disrupting agents, we inoculated conidia of different strains on CM medium containing sodium dodecyl sulfate (SDS, 0.005%) and Congo red (CR, 300 μg/mL). After 8 days, we detected no significant differences in the mycelial growth of all tested strains (Figure 2E). Taken together, these data suggest that NLP might play a role in *S. lycopersici* adaptation to osmotic and oxidative stresses, but not for maintenance of cell wall integrity.

### 3.4. NLP Is Required for Full Virulence of S. lycopersici in Tomato

To test the role of NLP in the virulence of *S. lycopersici* on tomato, we infected tomato leaves with the WT, ∆*nlp*, and *pACTIN:NLP* strains by placing droplets of conidial suspension (5 µL, 1 × 10^5^ conidia/mL) on the surfaces of the leaves. When we measured the resulting diseased areas at 5 days post inoculation (dpi), we found that, compared to those on the WT leaves, the disease lesions on the ∆*nlp* mutant leaves were significantly smaller, while those on *pACTIN:NLP* strains were significantly larger (Figure 3A,B and Appendix A). In addition, we quantified the mycelium biomass of the different strains at the infection sites and found that the mycelia of the ∆*nlp* mutant at the diseased sites were lower than those of the WT at 5 dpi (Figure 3C). The results suggested that the *NLP* gene deletion reduced *S. lycopersici* virulence. Overall, these data indicate that NLP is an important virulence factor in tomato grey leaf spot disease.

### 3.5. NLP Suppresses S. lycopersici Infection-Induced ROS Production in Tomato Leaves

It has been proposed that NLP acts as a PAMP that could trigger PTI-like responses including ROS production in the host [40,41]. To test whether *S. lycopersici* NLP could affect host ROS production, we assayed the transcription of the tomato ROS synthesis genes *RESPIRATORY BURST OXIDASE HOMOLOG A* (*RbohA*) and *RbohB* in infected tomato plants by RT-qPCR [42]. Tomato leaves were infected by the WT, ∆*nlp*, and *pACTIN:NLP* strains, and expression levels of *RbohA* and *RbohB* were quantified as 12 hpi and 24 hpi. The results showed that the WT strain induced the transcription of *RbohA* and *RbohB* in tomato leaves. Compared to the WT strain, the ∆*nlp* strain triggered significantly higher expression of *RbohA* and *RbohB*, while the *pACTIN:NLP* strain caused lower expression of both genes (Figure 4A,B). To further confirm that host ROS production is affected by *S. lycopersici* NLP, we detected H_2_O_2_ and O_2_^−^ production in infected tomato leaves at 18 hpi by DAB and NBT staining. Our results showed that the ∆*nlp* mutant strain caused more H_2_O_2_ (Figure 4C,D) and O_2_^−^ (Figure 4E,F) accumulation than the WT. Accordingly, lower ROS levels were detected in leaves infected with the *pACTIN:NLP* strain vs. WT (Figure 4C–F). Overall, these results suggest that *S. lycopersici* NLP suppresses the transcription of host ROS synthesis genes and inhibits infection-induced ROS production in the host.

### 3.6. Expression of S. lycopersici NLP in Tomato Leads to Constitutive Plant Immune Responses and Reduced Growth

To test whether the *S. lycopersici* NLP protein could trigger immune responses in tomato, the *NLP* gene sequence was fused with GFP and driven by the cauliflower mosaic virus (CaMV) 35S promoter (*p35S:NLP*). The *p35S:NLP* construct was introduced into tomato by *Agrobacterium*-mediated gene transformation. Expression of the *NLP* gene in *p35S:NLP* transgenic tomato was quantified by RT-qPCR. Our results showed that transgenic *p35S:NLP* tomato plants highly expressed the fungal *NLP* gene (Figure 5A). To determine the effect of *p35S:NLP* on tomato immune responses, we incubated *p35S:NLP* and WT tomato seedlings on half-strength Murashige and Skoog (1/2 MS) medium in sterile conditions and assessed their immune responses after 2 weeks. RT-qPCR data showed that the *p35S:NLP* transgenic plants constitutively expressed both *PR-STH2* and *ERF.C3* (Figure 5B,C), two immune-responsive genes that play an important role in tomato resistance to *B. cinerea* [26]. Interestingly, overexpressing *S. lycopersici NLP* gene in tomato resulted in a phenotype of reduced growth: the fresh weight of 1-week-old *p35S:NLP* transgenic plants was significantly lower than that of WT plants (Figure 5D). In addition, the primary roots of *p35S:NLP* tomato lines were significantly shorter than the WT (Figure 5E). Similarly, when grown in the greenhouse, the *p35S:NLP* lines showed lower growth, as well as significantly shorter height and lower fresh weight, compared to the WT (Figure 5F–H). These data suggest that *S. lycopersici* NLP triggers host immunity and inhibits plant growth.

## 4. Discussion

*Stemphylium* spp. fungal pathogens are economically important. Many are necrotrophic pathogens that infect a wide spectrum of crops worldwide, including tomato [6], pepper [43], eggplant [44], onion [45], parsley [3], garlic [4], lettuce [5], and cotton [46]. However, their pathogenicity is poorly understood due to a lack of functional genomic studies. Reliable genetic transformation methods are a prerequisite to studying the molecular functions of genes. It has been reported that targeted gene replacement could be enhanced by CRISPR/Cas-derived RNA-guided nucleases (RGNs) in the rice blast fungus *Pyricularia oryzae* [37]. In this study, we employed targeted gene replacement enhanced by RGN and established a gene transformation method by which we generated *S. lycopersici NLP* knockout mutants and overexpression strains. Using these genetically modified strains, we showed that NLP is a key virulence factor in tomato gray leaf spot disease caused by *S. lycopersici*.

Nep1 (necrosis- and ethylene-inducing peptide 1) was first identified as a 24-kDa elicitor protein from the fungal pathogen *Fusarium oxysporum* f. sp. *erythroxyli* that induces necrosis and ethylene production in the leaves of *Erythroxylum coca* and other dicots [47]. The *Nep1* gene from *F. oxysporum* was cloned, after which Nep1-like proteins (NLPs) were identified in other plant pathogens [48]. NLPs are taxonomically widespread in fungi, oomycetes, and bacteria [49,50]. The NLP family exhibits functional diversity, as demonstrated by the genetic manipulation of NLPs in many microorganisms including *F. oxysporum* [15], *B. cinerea* [17], *Magnaporthe oryzae* [51], *B. elliptica* [52], and *Diplodia seriata* [53].

Previous studies have suggested that NLPs can be either toxic or non-toxic to host plants [54]. For instance, Nep1 from *F. oxysporum* shows herbicidal potential against many dicot weed species [55]. The tomato-pathogenic *Verticillium dahliae* strain contains seven NLPs, of which only NLP1 and NLP2 induce plant cell death [56]. NLP1 also affects *V. dahliae* vegetative growth and conidiospore production [50,56]. It has been shown that toxic NLPs act as both phytotoxins and PAMPs. The toxic NLPs could induce necrosis, ROS and ethylene production, transcription of genes encoding pathogenesis-related proteins (PRs), and callose deposition [18,57]. In contrast, the non-toxic NLPs do not induce necrosis but are recognized by the host immune system and induce immune responses [58]. Thus, NLPs can be recognized by host receptors and trigger a PTI response. 

We identified a single copy of *NLP* in *S. lycopersici* based on the drafted genome. As *S. lycopersici* infected tomato plants, *NLP* expression was upregulated. Targeted gene disruption of the *NLP* gene significantly inhibited *S. lycopersici* virulence and conidiophore formation. We found that in *S. lycopersici*, NLP also plays positive roles in osmotic and oxidative stress adaptation. It is still unclear how NLPs regulate fungal development and stress adaptation. Large scale functional genomics studies in *S. lycopersici* are needed to further understand how virulence factors affect fungal development. Based on the disease assays we conducted in tomato, *S. lycopersici* NLP promotes necrotic lesion formation and immune-responsive gene expression in infected plants, and is thus toxic to the plant host. Interestingly, our results showed that *S. lycopersici* NLP suppresses host ROS production (Figure 4). The effect of NLP on ROS production in the host could inhibit plant immunity and facilitate *S. lycopersici* virulence. However, no receptors of any pathogen NLPs have so far been reported in tomato. It remains unclear how NLP could suppress ROS production and trigger immune responses in tomato.

It is commonly accepted that plant defense responses are energetically costly and typically lead to plant growth inhibition [59]. A previous study has shown that expressing a non-cytotoxic NLP from the biotrophic downy mildew pathogen *Hyaloperonospora arabidopsidis*, HaNLP3, in Arabidopsis triggered resistance to *H. arabidopsidis* and inhibited plant growth [60]. Similarly, in this study, we found that overexpression of the *S. lycopersici NLP* gene in tomato resulted in upregulation of immune-responsive genes and reduced plant growth (Figure 5). Because plants balance defense and growth, it has been a great challenge to improve both crop defense and yield. The tomato transgenic lines showed strong growth and defense phenotype, which makes this a powerful genetic tool to further study the mechanism of how plants balance defense and growth. 

In conclusion, our work represents the first comprehensive functional genomic study in the non-model plant pathogen *S. lycopersici*. Using gene manipulation methods, we generated targeted gene disruption mutants and ectopic overexpression lines of *NLP* in *S. lycopersici*, which allowed us to investigate its role in pathogen virulence and development. Our data suggest that NLP might contribute to the virulence of *S. lycopersici* by suppressing ROS-mediated immunity in tomato. Large-scale functional genomic studies are required to further dissect the pathogenicity of *Stemphylium* spp. 

## Figures and Tables

**Figure 1 jof-08-00518-f001:**
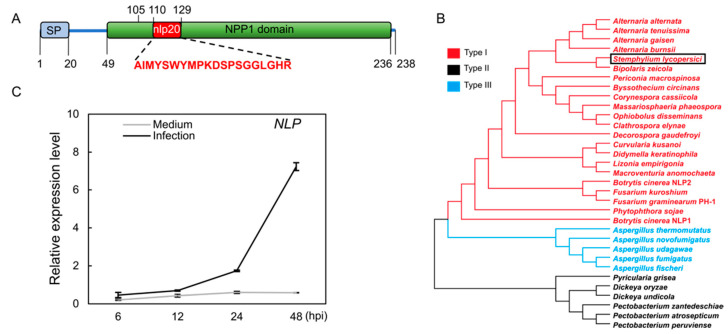
Characterization of a type I Nep1-like protein (NLP) from *S. lycopersici*. (**A**) Schematic diagram of the NLP protein of gray leaf spot. The full sequence of the NLP protein of gray leaf spot was used to predict and analyze the conserved structural domains. SP denotes signal peptide. NPP1 denotes Necrosis-Inducing Phytophthora Protein 1 characteristic domain. nlp20 represents the conserved pattern of 20 amino acids. (**B**) Phylogenetic tree built using three types of NLP proteins with maximum likelihood method. Red, black, and blue represent the type I, type II, and type III NLPs, respectively. The black box highlights the NLP of *S. lycopersici*. (**C**) Expression of the *NLP* gene during pathogen infection. The expression of the *NLP* gene was measured at 6, 12, 24, and 48 h post-inoculation (hpi) on tomato leaves and in CM liquid medium (negative control). The *S. lycopersici ACTIN* gene was used as an internal reference.

**Figure 2 jof-08-00518-f002:**
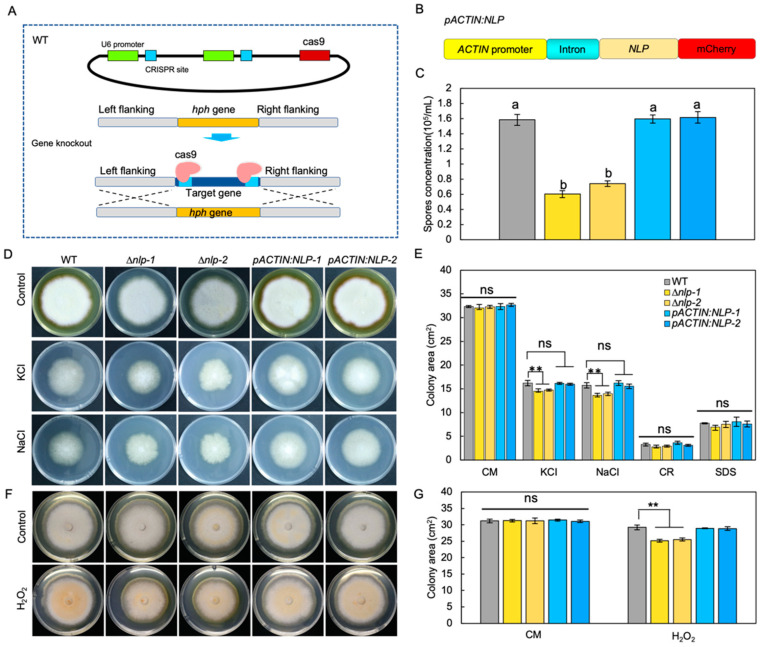
NLP affects the adaptation of *S. lycopersici* to ionic and oxidative stress as well as its asexual reproduction. (**A**) Schematic diagram showing targeted replacement of *NLP* enhanced by CRISPR/Cas9. (**B**) Schematic diagram showing the *S. lycopersici NLP* gene overexpression construct. The mCherry-tagged *NLP* gene was driven by the promoter and first intron of *ACTIN*. (**C**) Effect of NLP on conidia production. Strains were inoculated on sporulation medium. All spores were collected and suspended in liquid medium. The conidia production was assessed by measuring spore concentration. Lowercase of a and b denotes significant difference among multiple groups (*p* < 0.05) by Duncan’s new multiple range test. (**D**) Effect of NLP on the adaption of *S. lycopersici* to ionic stress and the cell wall disturbing agents sodium dodecyl sulfate (SDS, 0.005%) and Congo Red (CR, 300 μg/mL). Conidia of the wild-type (WT), ∆*nlp-1*, ∆*nlp-2*, *pACTIN:NLP-1*, and *pACTIN:NLP-2* strains were inoculated on CM medium containing NaCl (1 M) or KCl (1 M) and incubated at 25 °C for 8 days. CM medium without stress agents was used as the negative control. (**E**) Mycelial growth quantification of strains under ionic stress. (**F**) Effect of NLP on the adaption of *S. lycopersici* to oxidative stress. Fungal cakes (5 mm in diameter) of WT, ∆*nlp-1*, ∆*nlp-2*, *pACTIN:NLP-1*, and *pACTIN:NLP-2* strains were inoculated on CM plates containing H_2_O_2_ (20 mM) and incubated at 25 °C for 8 days. CM medium without H_2_O_2_ was used as the negative control. (**G**) Mycelial growth quantification of strains growing under oxidative stress. ** denotes very significant difference (*p* < 0.01, Student’s *t* test); 8 denotes significant difference (*p* < 0.05); ns denotes no significant differences.

**Figure 3 jof-08-00518-f003:**
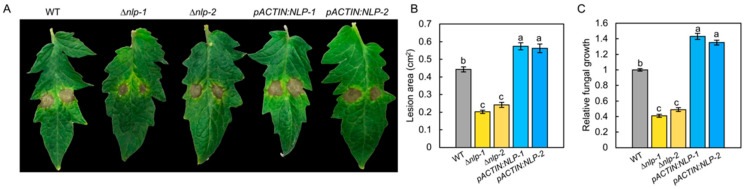
NLP is a key virulence factor of *S. lycopersici* during infection on tomato leaves. (**A**) Infected tomato leaves of the WT, ∆*nlp*, and overexpression strains at 5 days post inoculation (dpi). (**B**) Lesion area of leaves of tomato cultivar M82 resulting from infection by WT, ∆*nlp*, and overexpression strains. (**C**) Fungal biomass of WT, ∆*nlp,* and overexpression strains on infected tomato leaves. The relative fungal growth was measured by RNA-based RT-qPCR using the threshold cycle value (*C_T_*) of *S. lycopersici ACTIN* gene (locus_tag:TW65_02246) against the *C_T_* of tomato *ACTIN2* gene (Solyc11g005330). Conidia of different strains were inoculated on the surface of tomato leaves, and mycelium biomass was measured at 5 days post inoculation (dpi). The assay was performed on intact plants, of which the infected leaves were detached for imaging. Lowercase of a, b, and c denotes significant difference among multiple groups (*p* < 0.05) by Duncan’s new multiple range test.

**Figure 4 jof-08-00518-f004:**
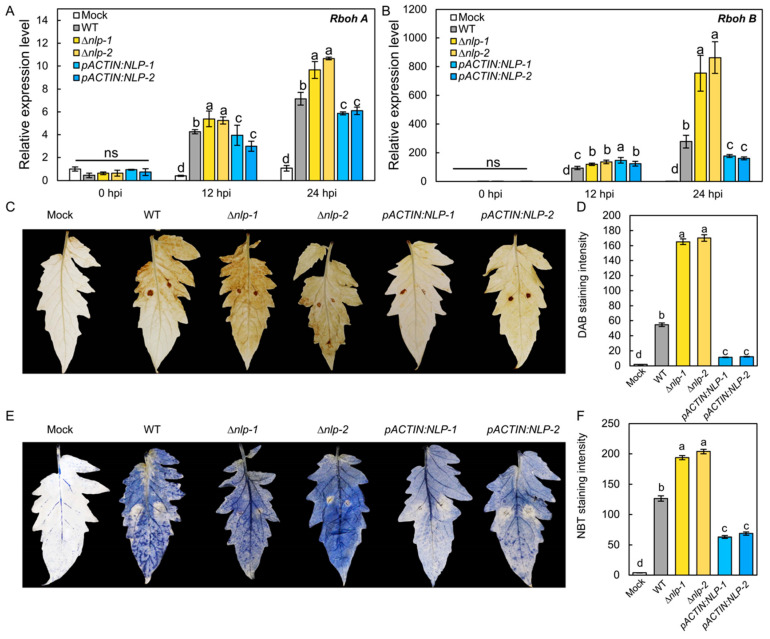
NLP inhibits ROS production in tomato induced by *S. lycopersici* infection. (**A**,**B**) Relative expression levels of tomato *RbohA* (**A**) and *RbohB* (**B**) gene after infection by the WT, ∆*nlp*, and overexpression *S. lycopersici* strains. (**C**) DAB staining of tomato leaves showing H_2_O_2_ production induced by the WT, ∆*nlp*, and overexpression *S. lycopersici* strains. (**D**) Quantification of DAB staining. (**E**) NBT staining of tomato leaves showing O_2_^−^ production induced by the WT, ∆*nlp*, and overexpression strains. (**F**) Quantification of NBT staining. The RGB values of images were converted into 16-bit grayscale, which were quantified by ImageJ with the grayscale statistics method. a, b, c, and d designate statistically significant differences determined using the DPS software. Lowercase of a, b, and c denotes significant difference among multiple groups (*p* < 0.05) by Duncan’s new multiple range test. ns denotes no significant differences.

**Figure 5 jof-08-00518-f005:**
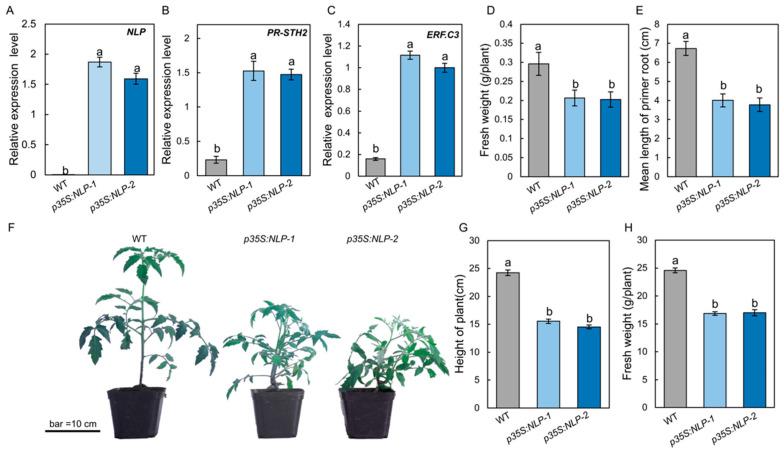
Expression of *S. lycopersici NLP* gene in tomato leads to constitutive expression of immunity genes and plant growth reduction. (**A**) RT-qPCR results showing the *NLP* gene expression in the leaves of *NLP*-overexpressing transgenic tomato lines. GFP-tagged NLP was driven by the 35S promoter (*p35S:NLP*). (**B**,**C**) Expression of the immune-responsive genes *PR-STH2* (**B**) and *ERF.C3* (**C**) in the *p35S:NLP* tomato lines. Tomato plants were incubated under sterile conditions for 2 weeks prior to RNA extraction and RT-qPCR analysis. (**D**,**E**) Fresh weight (**D**) and root length (**E**) of *p35S:NLP* tomato plants. Two-week-old tomato plants grown in growth container under sterile conditions were weighed (g per plant) and their root lengths were measured. (**F**–**H**) Growth phenotype (**F**), height (**G**), and fresh weight (**H**) of *p35S:NLP* tomato plants. Plants were grown in the greenhouse for 5 weeks. The scale bar indicates 10 cm. Lowercase of a and b denotes significant difference among multiple groups (*p* < 0.05) by Duncan’s new multiple range test.

## Data Availability

Not applicable.

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
