# Peer review of "Stemphylium lycopersici Nep1-like Protein (NLP) Is a Key Virulence Factor in Tomato Gray Leaf Spot Disease"

_jof, 2022, doi:10.3390/jof8050518_

Round 1

Reviewer 1 Report

Review of the manuscript entitled “Stemphylium lycopersici Nep1-Like Protein (NLP) Is a Key 2 Pathogenic Factor in Tomato Grey Leaf Spot Disease.”

The authors did a functional characterization of the Stemphylium lycopersici Nep1-Like Protein (NLP) and their role in pathogenicity. I have some major and few minor comments.

Major Comments:

Figure 1B:

The phylogenetic tree image in the Figure A is not a good quality/publication quality image. I would like to request the authors to change the image for figure 1 and include a high resolution image for the phylogenetic tree.

Figure S2A. Confirmation of NLP knockout and overexpression S. lycopersici strains

The DNA gel picture for confirmatory PCR to delete the NLP gene is again not a publication quality image. Please do another PCR and get a good gel picture to improve the quality of the figures.

Materials and method section:

Bioinformatics analysis and phylogenetic assay

The authors should mention the phylogenetic analysis of the NLP protein in more details, like the full description of the tree is it a maximum likelihood tree or a neighbor joining tree or a minimum evolutionary tree.

Genetic transformation of S. lycopersici

The authors did not mention how the protoplast transformation was done, was it done by chemical transformation or electroporation method. Although the authors cited the paper (Ref 30) but I still think it is better to mention the full protocol for the readers.

Minor Comments:

  1. It is very difficult to understand sometimes whether the authors are writing about the gene or the protein. Because in some cases the NLP is written in Italics to mention it as a gene but other places it is written in normal font. The authors can be more precise when they are mention about it in the manuscript.
  2. Did the authors tried to the plant infection with any other variety of tomato plants and see the same defect in the NLP mutants?

Reviewer 2 Report

The manuscript titled Stemphylium lycopersici Nep1-Like Protein (NLP) Is a Key Pathogenic Factor in Tomato Gray Leaf Spot Disease is a comprehensive research paper. The authors use various genetic and biological techniques, and the results are clearly presented.
The authors perform functional genomics studies satisfactorily, with their corresponding controls.
However, there is one point that casts doubt on the work carried out: Authors used an isolates identified as S. lycopersici based on ITS1-4 sequences. However, it is incorrect due to different species of Stemphylium have identical ITS sequences (See Câmara et al. 2017 (https://doi.org/10.1080/15572536.2003.11833194); Inderbitzin et al. 2005 (https:// doi.org/10.1073/pnas.0501918102), Inderbitzin et al. 2009 (https://doi.org/10.3852/08-071)). Thus, the authors cannot define the Stemphylium species they worked with due to the lack of other molecular markers. This is a serious mistake.

In line with this, are two methodologies that generate great doubts about the results obtained:

  1. In materials and methods authors described that WT isolate was transferred weekly to fresh medium. However, studies described that subcultures can modified conidiation, virulence and pathogenicity of fungi (See Ansari and Butt 2010 (doi:10.1111/j.1365-2672.2011.04994.x); Wang and Lin 2012 (doi:10.1371/journal.ppat.1003027); Fang et al. 2021 (https://doi.org/10.1371/journal.ppat.1009769)). So, how can the authors affirm that the observed changes are due to the variables studied and not to changes resulting from the subcultures?
  2. To test the effect to oxidative stress mutant strains were inoculated on CM plates containing H2O2 (20 mM) and incubated at 25 °C for 8 days. Hydrogen peroxide is an oxidizing agent and very unstable. So, how can the authors know its concentration in the culture medium, which is also incubated at 25? In addition, how do you know that I do not oxidize components of the culture medium? I think this assay needs to be revised.

Minor revision

  1. NLP is a pathogenic or a virulence factor? When NLP is knoked, virulence is minor but the isolates are pathogenic. The authors use both terms indiscriminately.
  2. On line 134…putative transformants were first tested by PCR and further confirmed by gene sequencing….detail please PCR and sequencing target.

  3. Discussion should be improved. In the present form, authors list the major findings. 

Reviewer 3 Report

The manuscript describes the characterization of a so called NEP1-like Protein (NLP) in the tomato pathogen Stemphylium lycopersici. NLP proteins are wide spreads amongst plant pathogenic fungi and oomycetes.

The paper is well written and abounding with correct English. Some of the results, however, are not fully documented. In particular, the data concerning the gene disruption are not sufficient to fully evaluate how the experiment was performed.

Most importantly, the interpretation of data is not always justified.

The data shown in Figure 2 for example were not interpreted correctly. A error bar with a star does not say anything concerning the biological relevance of quantitative data.

The data in Figure 3 show that there is little if any effect of NLP on virulence.

The data in Figure 5 confirms the biological effects of NEP expression in Arabidopsis.

In summary a well written paper with very limited scientific results that suffers from over interpretion of several pieces of data.

Specific comments:

Line 74: this would be a good opportunity to note how similar Stemphylium sequences are to those of other, better studies fungi, like Alternaria.

Line114: The relevant info was not provided (table S1). I need to see this first. It remains unclear how the gene replacement was done.

Line 130 please provide the PCR and sequencing data.

Line 118 which NLP gene was used?

Line 234: How was relative expression determined (relative to what?)

Line 258: it remains unclear how this was done. Please provide all relevant information, including DNA Sequences and PCR data.

Figure 1: define NPP1 domain.

Figure 2: The phenotypic effects are so

Figure 2S : The PCR is not conclusive. The authors need to show a PCR product for the deletion strain, not the absence of the WT allele which could just reflect a bad PCR !

Line 303: A reduction in conidiation by a factor of 2 is no evidence for an important role in conidiation. This is an overinterpretation.

Line 323: again the effect is an overinterpretation of the biological effect.

Figure 3: The data are over interpreted because mycelium length is so hard to measure that a factor of <2 cannot be interpreted. The authors may consider to measure DNA amount of the pathogen inside the lesions as a more reliable measure for virulence. The authors must also analyse intact plants not only detached leaves.

Round 2

Reviewer 2 Report

The work entitled “Stemphylium lycopersici Nep1-Like Protein (NLP) Is a Key 2 Virulence Factor in Tomato Gray Leaf Spot Disease” was greatly improved by the authors, according to the suggestions. It is a complete and complex genomic and functional study work of an important disease that affects tomato cultivars. I consider that the work is a first approximation to known the interaction between tomato-S. lycopersici. Based on these points, I suggest this research as acceptable to publication on JoF after minor revisions.

Minor revisions

  1. Authors use “type” and “Type” along the manuscript. I suggest unified the nomenclature (for type).
  2. Similarly, authors use NIPP. I suggest change “necrosis-inducing Phytophthora protein” by “Necrosis-Inducing Phytophthora Protein” along the manuscript.
  3. Authors should be include the bibliography used for oxidative stress assay.
  4. Authors should review the citation format.

Author Response

The work entitled “Stemphylium lycopersici Nep1-Like Protein (NLP) Is a Key 2 Virulence Factor in Tomato Gray Leaf Spot Disease” was greatly improved by the authors, according to the suggestions. It is a complete and complex genomic and functional study work of an important disease that affects tomato cultivars. I consider that the work is a first approximation to known the interaction between tomato-S. lycopersici. Based on these points, I suggest this research as acceptable to publication on JoF after minor revisions.

Response: We thank the reviewer for the high evaluation of our work!

Minor revisions

1. Authors use “type” and “Type” along the manuscript. I suggest unified the nomenclature (for type).

Response: Thanks this reviewer for pointing out this issue. We replaced "Type" with "type" in the manuscript.

2. Similarly, authors use NIPP. I suggest change “necrosis-inducing Phytophthora protein” by “Necrosis-Inducing Phytophthora Protein” along the manuscript.

Response: Thanks this reviewer for pointing out this issue. We replaced  “necrosis-inducing Phytophthora protein” with Necrosis-Inducing Phytophthora Protein” in the manuscript.

3. Authors should be include the bibliography used for oxidative stress assay.  

Response: Thanks this reviewer for pointing out this issue. We have added a reference corresponding to oxidative stress assay.

4. Authors should review the citation format.

Response: Thanks this reviewer for pointing out this issue. We have updated the references

Reviewer 3 Report

The authors have improved their ms.

However the critical experiment in Figure 3 and the corresponding Methods section contradict each other and it remains unclear how the fungal biomass was determined and whether not not the experiments were performed on intact leaves. 

This must be changed and the title is misleading as it suggests stronger effects  than the actual data show.

Author Response

The authors have improved their ms.

However the critical experiment in Figure 3 and the corresponding Methods section contradict each other and it remains unclear how the fungal biomass was determined and whether not not the experiments were performed on intact leaves. 

Response: Thanks this reviewer for pointing out this issue. We have added information in the Materials and Methods section 2.9 and updated the figure legend.

This must be changed and the title is misleading as it suggests stronger effects  than the actual data show.

Response:Thanks this reviewer for pointing out this issue. We have changed the title to “NLP is a key virulence factor of S. lycopersici during infection on tomato leaves”.